# Comprehension of computer code relies primarily on domain-general executive brain regions

**Anna A Ivanova[1,2]\*, Shashank Srikant[3], Yotaro Sueoka[1,2], Hope H Kean[1,2], Riva Dhamala[4], Una-May O'Reilly[3], Marina U Bers[4], Evelina Fedorenko[1,2]\***

[1]Department of Brain and Cognitive Sciences, Massachusetts Institute of Technology, Cambridge, United States; [2]McGovern Institute for Brain Research, Massachusetts Institute of Technology, Cambridge, United States; [3]Computer Science and Artificial Intelligence Laboratory, Massachusetts Institute of Technology, Cambridge, United States; [4]Eliot-Pearson Department of Child Study and Human Development, Tufts University, Medford, United States

**Abstract** Computer programming is a novel cognitive tool that has transformed modern society. What cognitive and neural mechanisms support this skill? Here, we used functional magnetic resonance imaging to investigate two candidate brain systems: the multiple demand (MD) system, typically recruited during math, logic, problem solving, and executive tasks, and the language system, typically recruited during linguistic processing. We examined MD and language system responses to code written in Python, a text-based programming language (Experiment 1) and in ScratchJr, a graphical programming language (Experiment 2); for both, we contrasted responses to code problems with responses to content-matched sentence problems. We found that the MD system exhibited strong bilateral responses to code in both experiments, whereas the language system responded strongly to sentence problems, but weakly or not at all to code problems. Thus, the MD system supports the use of novel cognitive tools even when the input is structurally similar to natural language.

**\*For correspondence:**
annaiv@mit.edu (AAI);
evelina9@mit.edu (EF)

**Competing interests:** The authors declare that no competing interests exist.

## Introduction

The human mind is endowed with a remarkable ability to support novel cognitive skills, such as reading, writing, map-based navigation, mathematical reasoning, and scientific logic. Recently, humanity has invented another powerful cognitive tool: computer programming. The ability to flexibly instruct programmable machines has led to a rapid technological transformation of communities across the world (*Ensmenger, 2012*); however, little is known about the cognitive and neural systems that underlie computer programming skills.

Here, we investigate which neural systems support one critical aspect of computer programming: computer code comprehension. By code comprehension, we refer to a set of cognitive processes that allow programmers to interpret individual program tokens (such as keywords, variables, and function names), combine them to extract the meaning of program statements, and, finally, combine the statements into a mental representation of the entire program. It is important to note that code comprehension may be cognitively and neurally separable from cognitive operations required to process program content, that is, the actual operations described by code. For instance, to predict the output of the program that sums the first three elements of an array, the programmer should identify the relevant elements and then mentally perform the summation. Most of the time, processing program content recruits a range of cognitive processes known as computational thinking (*Wing, 2006*; *Wing, 2011*), which include algorithm identification, pattern generalization/

abstraction, and recursive reasoning (e.g., *Kao, 2010*). These cognitive operations are notably different from code comprehension per se and may not require programming knowledge at all (*Guzdial, 2008*). Thus, research studies where people read computer programs should account for the fact that interpreting a computer program involves two separate cognitive phenomena: processing computer code that comprises the program (i.e., code comprehension) and mentally simulating the procedures described in the program (i.e., processing problem content).

Given that code comprehension is a novel cognitive tool, typically acquired in late childhood or in adulthood, we expect it to draw on preexisting cognitive systems. However, the question of which cognitive processes support code comprehension is nontrivial. Unlike some cognitive inventions that are primarily linked to a single cognitive domain (e.g., reading/writing building on spoken language), code comprehension plausibly bears parallels to multiple distinct cognitive systems. First, it may rely on domain-general executive resources, including working memory and cognitive control (*Bergersen and Gustafsson, 2011*; *Nakagawa et al., 2014*; *Nakamura et al., 2003*). In addition, it may draw on the cognitive systems associated with math and logic (*McNamara, 1967*; *Papert, 1972*), in line with the traditional construal of coding as problem-solving (*Dalbey and Linn, 1985*; *Ormerod, 1990*; *Pea and Kurland, 1984*; *Pennington and Grabowski, 1990*). Finally, code comprehension may rely on the system that supports comprehension of natural languages (*Fedorenko et al., 2019*; *Murnane, 1993*; *Papert, 1993*). Like natural language, computer code makes heavy use of hierarchical structures (e.g., loops, conditionals, and recursive statements), and, like language, it can convey an unlimited amount of meaningful information (e.g., describing objects or action sequences). These similarities could, in principle, make the language circuits well suited for processing computer code.

Neuroimaging research is well positioned to disentangle the relationship between code comprehension and other cognitive domains. Many cognitive processes are known to evoke activity in specific brain regions/networks: thus, observing activity for the task of interest in a particular region or network with a known function can indicate which cognitive processes are likely engaged in that task (*Mather et al., 2013*). Prior research (*Assem et al., 2020*; *Duncan, 2010*; *Duncan, 2013*; *Duncan and Owen, 2000*) has shown that executive processes – such as attention, working memory, and cognitive control – recruit a set of bilateral frontal and parietal brain regions collectively known as the multiple demand (MD) system. If code comprehension primarily relies on domain-general executive processes, we expect to observe code-evoked responses within the MD system, distributed across both hemispheres. Math and logic also evoke responses within the MD system (*Fedorenko et al., 2013*), although this activity tends to be left-lateralized (*Amalric and Dehaene, 2016*; *Amalric and Dehaene, 2019*; *Goel and Dolan, 2001*; *Micheloyannis et al., 2005*; *Monti et al., 2007*; *Monti et al., 2009*; *Pinel and Dehaene, 2010*; *Prabhakaran et al., 1997*; *Reverberi et al., 2009*). If code comprehension draws on the same mechanisms as math and logic, we expect to observe left-lateralized activity within the MD system. Finally, comprehension of natural language recruits a set of left frontal and temporal brain regions known as the language system (e.g., *Fedorenko and Thompson-Schill, 2014*). These regions respond robustly to linguistic input, both visual and auditory (*Deniz et al., 2019*; *Fedorenko et al., 2010*; *Nakai et al., 2020*; *Regev et al., 2013*; *Scott et al., 2017*). However, they show little or no response to tasks in non-linguistic domains, such as executive functions, math, logic, music, action observation, and non-linguistic communicative signals, like gestures (*Fedorenko et al., 2011*; *Jouravlev et al., 2019*; *Monti et al., 2009*; *Monti et al., 2012*; *Pritchett et al., 2018*; see *Fedorenko and Blank, 2020*, for a review). If code comprehension relies on the same circuits that map form to meaning in natural language, we expect to see activity within the language system.

Evidence from prior neuroimaging investigations of code comprehension is inconclusive. Existing studies have provided some evidence for left-lateralized activity in regions that roughly correspond to the language system (*Siegmund et al., 2014*; *Siegmund et al., 2017*), as well as some evidence for the engagement of frontal and parietal regions resembling the MD system (*Floyd et al., 2017*; *Huang et al., 2019*; *Siegmund et al., 2014*; *Siegmund et al., 2017*). However, none of these prior studies sought to explicitly distinguish code comprehension from other programming-related processes, and none of them provide quantitative evaluations of putative shared responses to code and other tasks, such as working memory, math, or language (cf. *Liu et al., 2020*; see Discussion).

Here, we use functional magnetic resonance imaging (fMRI) to evaluate the role of the MD system and the language system in computer code comprehension. Three design features that were lacking

in earlier neuroimaging studies of programming allow us to evaluate the relative contributions of these two candidate systems. First, we contrast neural responses evoked by code problems with those evoked by content-matched sentence problems (*Figure 1A*); this comparison allows us to disentangle activity evoked by code comprehension from activity evoked by the underlying program content (which is matched across code and sentence problems).

Second, we use independent 'localizer' tasks (*Brett et al., 2002*; *Fedorenko et al., 2010*; *Saxe et al., 2006*) to identify our networks of interest: a working memory task to localize the MD system and a passive reading task to localize the language system (*Figure 1B*). The functional localization approach obviates the reliance on the much-criticized 'reverse inference' reasoning (*Poldrack, 2006*; *Poldrack, 2011*), whereby functions are inferred from coarse macro-anatomical landmarks. Instead, we can directly interpret code-evoked activity within functionally defined regions

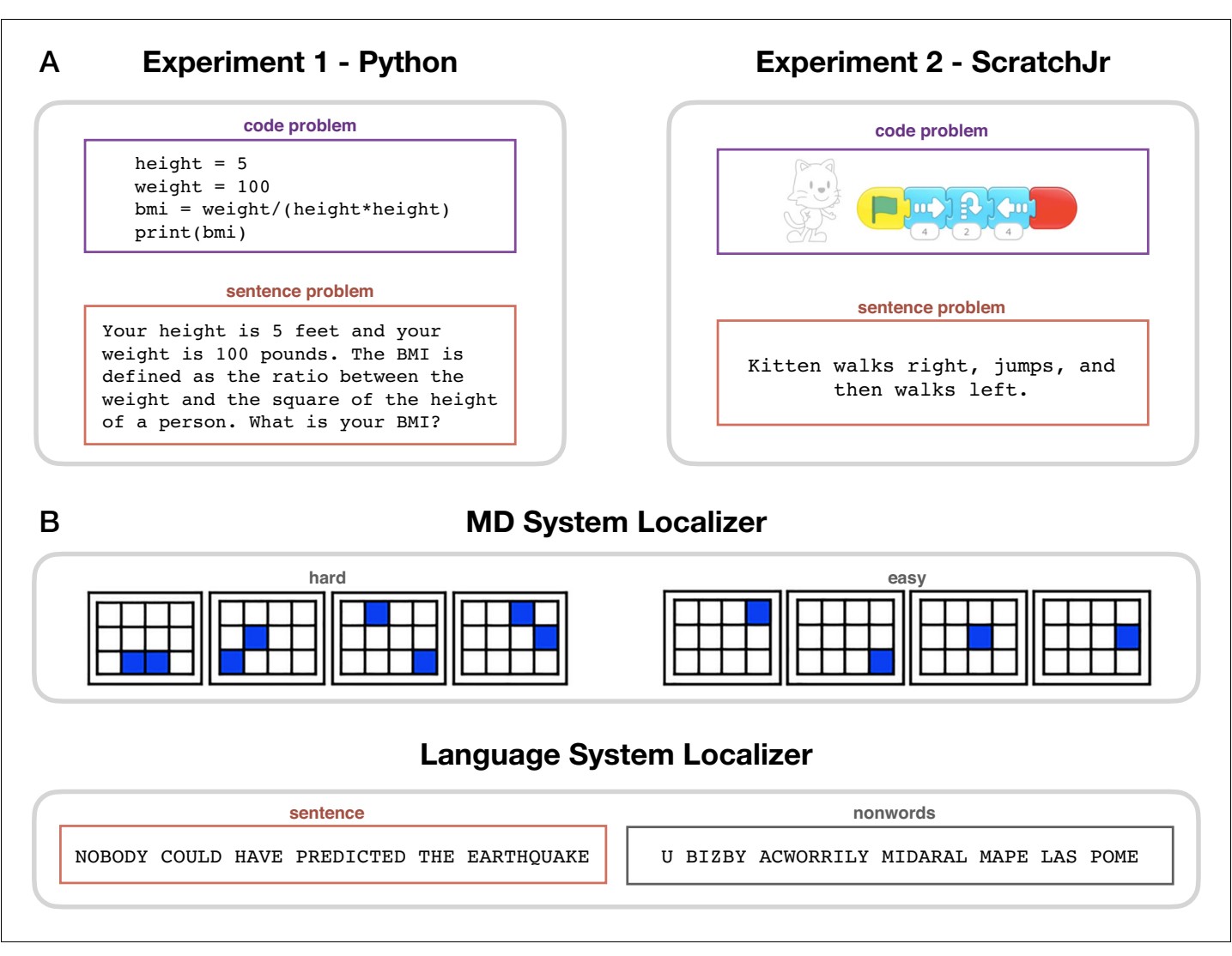

**Figure 1.** Experimental paradigms. (**A**) Main task. During code problem trials, participants were presented with snippets of code in Python (Experiment 1) or ScratchJr (Experiment 2); during sentence problem trials, they were presented with text problems that were matched in content with the code stimuli. Each participant saw either the code or the sentence version of any given problem. (**B**) Localizer tasks. The MD localizer (top) included a hard condition (memorizing positions of eight squares appearing two at a time) and an easy condition (memorizing positions of four squares appearing one at a time). The language localizer (bottom) included a sentence reading and a nonword reading condition, with the words/nonwords appearing one at a time.

The online version of this article includes the following figure supplement(s) for figure 1:

**Figure supplement 1.** Trial structure of the critical task.

of interest (*Mather et al., 2013*). In addition, localization of the MD and language networks is performed in individual participants, which is important given substantial variability in their precise locations across individuals (*Fedorenko and Blank, 2020*; *Shashidhara et al., 2019b*) and leads to higher sensitivity and functional resolution (*Nieto-Castañón and Fedorenko, 2012*).

Third, to draw general conclusions about code comprehension, we investigate two very different programming languages: Python, a popular general-purpose programming language, and ScratchJr, an introductory visual programming language for creating animations, designed for young children (*Bers and Resnick, 2015*). In the Python experiment, we further examine two problem types (math problems and string manipulation) and three basic types of program structure (sequential statements, *for* loops, and *if* statements). Comprehension of both Python and ScratchJr code requires retrieving the meaning of program tokens and combining them into statements, despite the fact that the visual features of the tokens in the two languages are very different (text vs. images). If a brain system is involved in code comprehension, we expect its response to generalize across programming languages and problem types, similar to how distinct natural languages in bilinguals and multilinguals draw on the same language regions (*Kroll et al., 2015*).

Taken together, these design features of our study allow us to draw precise and generalizable conclusions about the neural basis of code comprehension.

## Results

Participants performed a program comprehension task inside an MRI scanner. In each trial, participants, all proficient in the target programming language, read either a code problem or a content-matched sentence problem (*Figure 1A*) and were asked to predict the output. In Experiment 1 (24 participants, 15 women), code problems were written in Python, a general-purpose text-based programming language (*Sanner, 1999*). In Experiment 2 (19 participants, 12 women), code problems were written in ScratchJr, an introductory graphical programming language developed for children aged 5–7 (*Bers, 2018*). Both experiments were conducted with adults to facilitate result comparison. Good behavioral performance confirmed that participants were proficient in the relevant programming language and engaged with the task (Python: 99.6% response rate, 85% accuracy on code problems; ScratchJr: 98.6% response rate, 79% accuracy on code problems; see *Figure 2—figure supplement 1* for detailed behavioral results). Participants additionally performed two functional localizer tasks: a hard vs. easy spatial working memory task, used to define the MD system, and a sentence vs. nonword reading task, used to define the language system (*Figure 1B*; see Materials and methods for details).

We then contrasted neural activity in the MD and language systems during code problem comprehension with activity during (a) sentence problem comprehension and (b) the nonword reading condition from the language localizer task. Sentence problem comprehension requires simulating the same operations as code problem comprehension (mathematical operations or string manipulation for Python, video simulation for ScratchJr), so contrasting code problems with sentence problems allows us to isolate neural responses evoked by code comprehension from responses evoked by processing problem content. Nonword reading elicits weak responses in both the language system and the MD system (in the language system, this response likely reflects low-level perceptual and/or phonological processing; in the MD system, it likely reflects the basic task demands associated with maintaining attention or reading pronounceable letter strings). Because the nonword response is much weaker than responses to the localizer conditions of interest (*Fedorenko et al., 2010*; *Mineroff et al., 2018*), nonword reading can serve as a control condition for both the MD and language systems, providing a more stringent baseline than simple fixation. Given the abundant evidence that the MD system and the language system are each strongly functionally interconnected (*Blank et al., 2014*; *Mineroff et al., 2018*; *Paunov et al., 2019*), we perform the key analyses at the system level.

### MD system exhibits robust and generalizable bilateral responses during code comprehension

We found strong bilateral responses to code problems within the MD system in both Experiments 1 and 2 (*Figures 2* and *3*). These responses were stronger than responses to both the sentence problem condition (Python: β = 1.03, p<0.001, ScratchJr: β = 1.38, p<0.001) and the control nonword

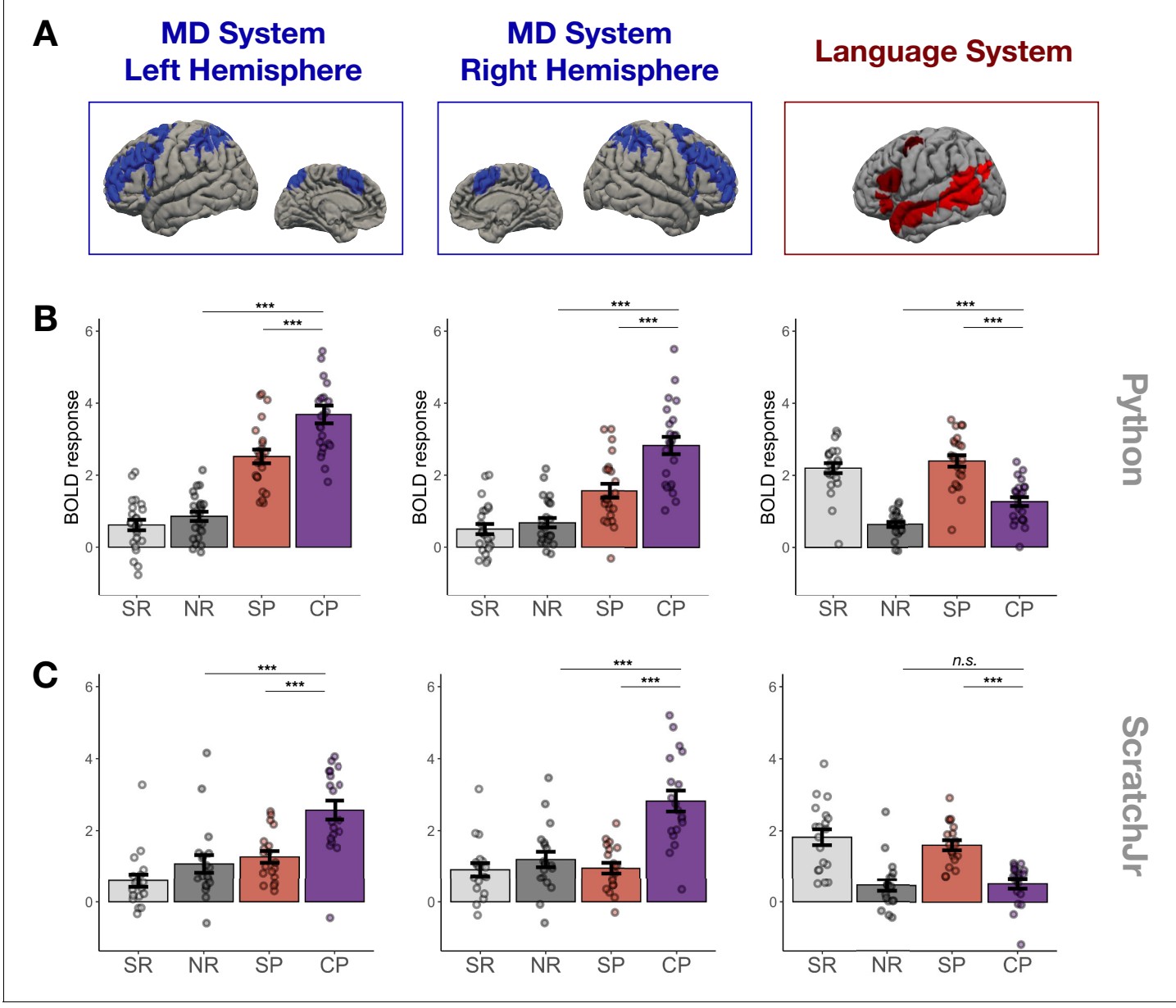

**Figure 2.** Main experimental results. (**A**) Candidate brain systems of interest. The areas shown represent the 'parcels' used to define the MD and language systems in individual participants (see Materials and methods and *Figure 3—figure supplement 1*). (**B, C**) Mean responses to the language localizer conditions (SR – sentence reading and NR – nonwords reading) and to the critical task (SP – sentence problems and CP – code problems) in systems of interest across programming languages (B – Python, C – ScratchJr). In the MD system, we see strong responses to code problems in both hemispheres and to both programming languages; the fact that this response is stronger than the response to content-matched sentence problems suggests that it reflects activity evoked by code comprehension per se rather than just activity evoked by problem content. In the language system, responses to code problems elicit a response that is substantially weaker than that elicited by sentence problems; further, only in Experiment 1 do we observe responses to code problems that are reliably stronger than responses to the language localizer control condition (nonword reading). Here and elsewhere, error bars show standard error of the mean across participants, and the dots show responses of individual participants.

The online version of this article includes the following figure supplement(s) for figure 2:

**Figure supplement 1.** Behavioral results.
**Figure supplement 2.** Random-effects group-level analysis of Experiment 1 data (Python, code problems > sentence problems contrast).
**Figure supplement 3.** Random-effects group-level analysis of Experiment 2 data (ScratchJr, code problems > sentence problems contrast).

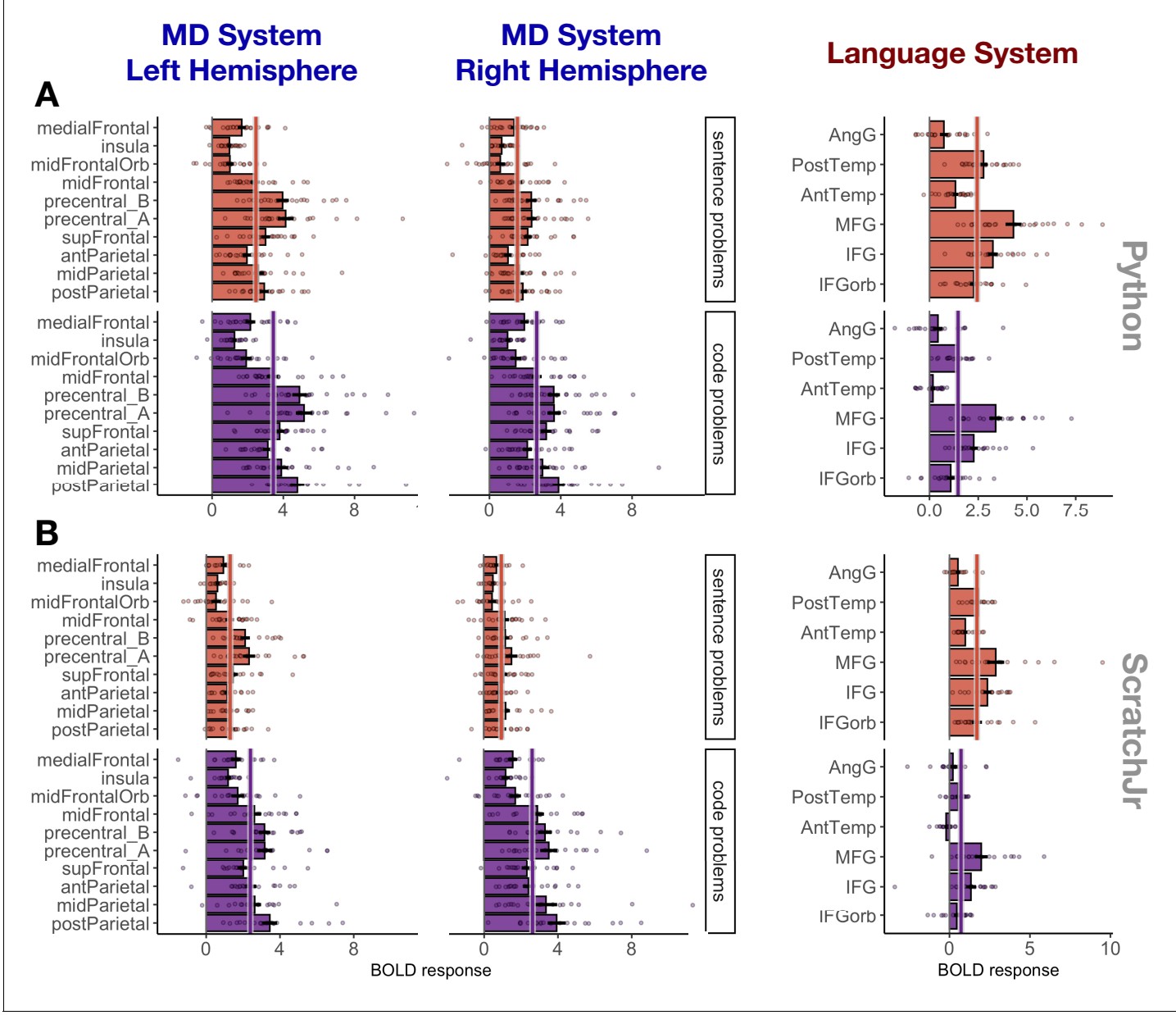

**Figure 3.** Responses to sentence problems (red) and code problems (purple) during Experiment 1 (Python; A) and Experiment 2 (ScratchJr; B) broken down by region within each system. Abbreviations: mid – middle, ant – anterior, post – posterior, orb – orbital, MFG – middle frontal gyrus, IFG – inferior frontal gyrus, temp – temporal lobe, AngG – angular gyrus, precentral_A – the dorsal portion of precentral gyrus, precentral_B – the ventral portion of precentral gyrus. A solid line through the bars in each subplot indicates the mean response across the fROIs in that plot.

The online version of this article includes the following figure supplement(s) for figure 3:

**Figure supplement 1.** The parcels in the two candidate brain systems of interest, multiple demand (MD) and language.

**Figure supplement 2.** ROI-level responses in the multiple demand system to the critical task (CP – code problems, SP – sentence problems) and the spatial working memory task (HardWM – hard working memory task, EasyWM – easy working memory task).

**Figure supplement 3.** Whole-brain group-constrained subject-specific (GSS) analysis (*Fedorenko et al., 2010*) based on data from Experiment 1 shows the absence of code-only brain regions.

**Figure supplement 4.** Whole-brain group-constrained subject-specific (GSS) analysis (*Fedorenko et al., 2010*) based on data from Experiment 2.

reading condition (Python: β = 2.17, p<0.001; ScratchJr: β = 1.23, p<0.001). The fact that code problems drove the MD system more strongly than content-matched sentence problems (despite the fact that sentence problems generally took longer to respond to; see *Figure 2—figure supplement 1*) demonstrates that the MD system responds to code comprehension specifically rather than simply being activated by the underlying problem content.

To further test the generalizability of MD responses, we capitalized on the fact that our Python stimuli systematically varied along two dimensions: (1) problem type (math problems vs. string manipulation) and (2) problem structure (sequential statements, *for* loops, *if* statements). Strong responses were observed in the MD system (*Figure 4A and B*) regardless of problem type (β = 3.02, p<0.001; no difference between problem types) and problem structure (β = 3.14, p<0.001; sequential problems evoked a slightly weaker response, β = −0.20, p=0.002). This analysis demonstrates that the responses were not driven by one particular type of problem or by mental operations related to the processing of a particular code structure.

We also tested whether MD responses to code showed a hemispheric bias similar to what is typically seen for math and logic problems (*Goel and Dolan, 2001*; *Micheloyannis et al., 2005*; *Monti et al., 2007*; *Monti et al., 2009*; *Pinel and Dehaene, 2010*; *Prabhakaran et al., 1997*; *Reverberi et al., 2009*). Neither Python nor ScratchJr problems showed a left-hemisphere bias for code comprehension. For Python, the size of the code problems > sentence problems effect did not interact with hemisphere (β = 0.11, p=0.46), even though the magnitude of responses to code problems as compared to nonword reading was stronger in the left hemisphere (β = 0.63, p<0.001). These results show that neural activity evoked by Python code comprehension was bilaterally distributed but that activity evoked by the underlying problem content was left-lateralized. For ScratchJr, the size of the code problems > sentence problems effect interacted with hemisphere, with stronger responses in the right hemisphere (β = 0.57, p=0.001), perhaps reflecting the bias of the right hemisphere toward visuo-spatial processing (*Corballis, 2003*; *Hugdahl, 2011*; *Sheremata et al., 2010*).

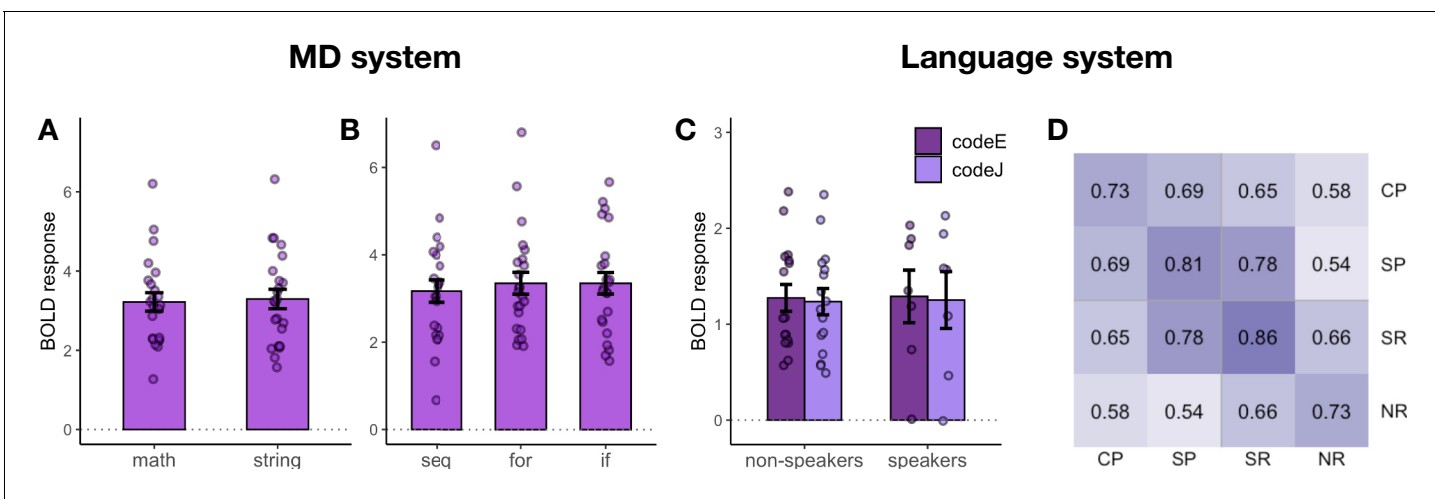

**Figure 4.** Follow-up analyses of responses to Python code problems. (**A**) MD system responses to math problems vs. string manipulation problems. (**B**) MD system responses to code with different structure (sequential vs. for loops vs. if statements). (**C**) Language system responses to code problems with English identifiers (codeE) and code problems with Japanese identifiers (codeJ) in participants with no knowledge of Japanese (non-speakers) and some knowledge of Japanese (speakers) (see the 'Language system responses...' section for details of this manipulation). (**D**) Spatial correlation analysis of voxel-wise responses within the language system during the main task (SP – sentence problems and CP – code problems) with the language localizer conditions (SR – sentence reading and NR – nonwords reading). Each cell shows a correlation between the activation patterns for each pair of conditions. Within-condition similarity is estimated by correlating activation patterns across independent runs.

The online version of this article includes the following figure supplement(s) for figure 4:

**Figure supplement 1.** Spatial correlation analysis of voxel responses within the MD system during the Python experiment (CP – code problems and SP – sentence problems) with the language localizer conditions for the same participants (SR – sentence reading and NR – nonword reading).

**Figure supplement 2.** The effect of programming expertise on code-specific response strength within the MD and language system in Experiment 1, Python (A, B) and Experiment 2, ScratchJr (C, D).

Follow-up analyses of activity in individual regions within the MD system demonstrated that 17 of the 20 MD fROIs (all except the fROIs located in left medial frontal cortex and in the left and right insula) responded significantly more strongly to Python code problems than to sentence problems (see *Supplementary file 1* for all fROI statistics). Responses to ScratchJr were significantly stronger than responses to sentence problems in 6 of the 10 left hemisphere MD fROIs (the effect was not significant in the fROIs in superior frontal gyrus, the dorsal part of the precentral gyrus, the medial frontal cortex, and the insula) and in 8 of the 10 right hemisphere MD fROIs (the effect was not significant in the fROIs in the medial frontal cortex and the insula; see *Supplementary file 1* for all fROI statistics). These analyses demonstrate that code processing is broadly distributed across the MD system rather than being localized to a particular region or to a small subset of regions.

Overall, we show that MD responses to code are strong, do not exclusively reflect responses to problem content, generalize across programming languages and problem types, and are observed across most MD fROIs.

## No MD fROIs are selective for code comprehension

To determine whether any fROIs were driven selectively (preferentially) by code problems relative to other cognitively demanding tasks, we contrasted individual fROI responses to code problems with responses to a hard working memory task from the MD localizer experiment. Three fROIs located in the left frontal lobe ('precentral_A', 'precentral_B', and 'midFrontal') exhibited stronger responses to Python code problems than to the hard working memory task ($\beta = 1.21$, $p<0.001$; $\beta = 1.89$, $p<0.001$; and $\beta = 0.79$, $p=0.011$, respectively; *Figure 3—figure supplement 2*). However, the magnitude of the code problems > sentence problems contrast in these regions ($\beta = 1.03, 0.95, 0.97$) was comparable to the average response magnitude across all MD fROIs (average $\beta = 1.03$), suggesting that the high response was caused by processing the underlying problem content rather than by code comprehension per se. Furthermore, neither these nor any other MD fROIs exhibited higher responses to ScratchJr code compared to the hard working memory task (in fact, the 'precentral_A' fROI did not even show a significant code problems > sentence problems effect). We conclude that code comprehension is broadly supported by the MD system (similar to, e.g., intuitive physical inference; *Fischer et al., 2016*), but no MD regions are functionally specialized to process computer code.

## Language system responses during code comprehension are weak and inconsistent

The responses to code problems within the language system (*Figures 2* and *3*) were weaker than responses to sentence problems in both experiments (Python: $\beta = 0.98$, $p<0.001$; ScratchJr: $\beta = 0.99$, $p<0.001$). Furthermore, although the responses to code problems were stronger than the responses to nonword reading for Python ($\beta = 0.78$, $p<0.001$), this was not the case for ScratchJr ($\beta = 0.15$, $p=0.29$), suggesting that the language system is not consistently engaged during computer code comprehension.

We further tested whether responses to Python code problems within the language system may be driven by the presence of English words. Our stimuli were constructed such that half of the Python problems contained meaningful identifier names, and in the other half, the English identifiers were replaced with their Japanese translations, making them semantically meaningless for non-speakers of Japanese. For this analysis, we divided our participants into two groups – those with no reported knowledge of Japanese (N = 18) and those with some knowledge of Japanese (N = 6) – and compared responses within their language regions to code problems with English vs. Japanese identifiers (*Figure 4C*). We found no effect of identifier language ($\beta = 0.03$, $p=0.84$), knowledge of Japanese ($\beta = 0.03$, $p=0.93$), or interaction between them ($\beta = 0.09$, $p=0.71$), indicating that the language system's response to Python code was not driven by the presence of semantically transparent identifiers. This result is somewhat surprising given the language system's strong sensitivity to word meanings (e.g., *Anderson et al., 2019*; *Binder et al., 2009*; *Fedorenko et al., 2010*; *Fedorenko et al., 2020*; *Pereira et al., 2018*). One possible explanation is that participants do not deeply engage with the words' meanings in these problems because these meanings are irrelevant to finding the correct solution.

Finally, we investigated whether the responses to Python code problems within the language system were driven by code comprehension specifically or rather by the underlying problem content. When examining responses in the MD system, we could easily disentangle the neural correlates of code comprehension vs. the processing of problem content using univariate analyses: the code problems > sentence problems contrast isolated code-comprehension-related processes, and the sentence problems > nonword reading contrast isolated responses to problem content. In the language system, however, the sentence problems > nonword reading response is additionally driven by language comprehension (unlike the MD system, which does not respond to linguistic input in the absence of task demands, as evidenced by its low responses during sentence reading; see also *Blank and Fedorenko, 2017*; *Diachek et al., 2020*). Thus, responses to Python code might be evoked both by problem content and by the language-like features of Python code. To determine the relative contributions of these two factors, we computed voxel-wise spatial correlations within and between the code problem and sentence problem conditions, as well as correlations between these conditions and the sentence/nonword reading conditions from the language localizer task (*Figure 4D*). We reasoned that if a system is driven by problem content, the activation patterns for code and sentence problems should be similar; in contrast, if a system is driven by code comprehension per se, the activation patterns for code and sentence problems should differ. We found that the activation patterns were highly correlated between the code and sentence problems (r = 0.69, p<0.001). These correlation values were higher than the correlations between code problems and sentence reading (0.69 vs. 0.65; p<0.001), although lower than the correlations within the code problem condition (0.69 vs. 0.73; p<0.001). The fact that code and sentence problem responses are correlated over and above code problem and sentence reading responses indicates that the language system is sensitive to the content of the stimulus rather than just the stimulus type (code vs. words). Moreover, similar to the MD system, problem content can account for a substantial portion of the response in the language regions (Δr = 0.04). Note that a similar spatial correlation analysis in the MD system mirrored the result of univariate analyses (*Figure 4—figure supplement 1*). Thus, in both MD and language systems, response to Python code is driven both by problem content and by code-specific responses.

Overall, we found that the language system responded to code problems written in Python but not in ScratchJr. Furthermore, Python responses were driven not only by code comprehension, but also by the processing of problem content. We conclude that successful comprehension of computer code can proceed without engaging the language network.

## No consistent evidence of code-responsive regions outside the MD/ language systems

To search for code-responsive regions that might fall outside the MD and language systems, we performed a whole-brain GSS analysis (*Fedorenko et al., 2010*). GSS analysis serves the same goal as the traditional random-effects voxel-wise analysis (*Holmes and Friston, 1998*) but accommodates inter-individual variability in the precise locations of functional regions, thus maximizing the likelihood of finding responsive regions (*Nieto-Castañón and Fedorenko, 2012*). We searched for areas of activation for the code problems > sentence problems contrast (separately for Python and ScratchJr) that were spatially similar across participants. We then examined the response of such regions to code and sentence problems (using an across-runs cross-validation procedure; e.g., *Nieto-Castañón and Fedorenko, 2012*), as well as to conditions from the two localizer experiments. In both experiments, the discovered regions spatially resembled the MD system (*Figure 3—figure supplements 3* and *4*). For Python, any region that responded to code also responded to the spatial working memory task (the MD localizer). In case of ScratchJr, some fROIs responded more strongly to code problems than to the spatial working memory task; these fROIs were located in early visual areas/ventral visual stream and therefore likely responded to low-level visual properties of ScratchJr code (which includes colorful icons, objects, etc.). The traditional random-effects group analyses revealed a similar activation pattern (*Figure 2—figure supplements 2* and *3*). These whole-brain analyses demonstrate that the MD system responds robustly and consistently to computer code, recapitulating the results of the fROI-based analyses (*Figures 2–4*), and show that fROI-based analyses did not miss any non-visual code-responsive or code-selective regions outside the boundaries of the MD system.

## Effect of proficiency on MD and language responses

We conducted an exploratory analysis to check whether engagement of the MD and/or language system in code comprehension varies with the level of programming expertise. We correlated responses within each system with independently obtained proficiency scores for Experiment 1 participants (see the paper's website for details: https://github.com/ALFA-group/neural-program-comprehension; *Ivanova and Srikant, 2020*; copy archived at swh:1:rev:616e893d05038-da620bdf9f2964bd3befba75dc5) and with in-scanner accuracy scores for Experiment 2 participants. No correlations were significant (see *Figure 4—figure supplement 2*). However, due to a relatively low number of participants (N = 24 and N = 19, respectively), these results should be interpreted with caution.

## Discussion

The ability to interpret computer code is a remarkable cognitive skill that bears parallels to diverse cognitive domains, including general executive functions, math, logic, and language. The fact that coding can be learned in adulthood suggests that it may rely on existing cognitive systems. Here, we tested the role of two candidate neural systems in computer code comprehension: the domain-general MD system (*Duncan, 2010*), which has been linked to diverse executive demands and implicated in math and logic (e.g., *Amalric and Dehaene, 2019*; *Goel, 2007*; *Monti et al., 2007*; *Monti et al., 2009*), and the language-selective system (*Fedorenko et al., 2011*), which has been linked to lexical and combinatorial linguistic processes (e.g., *Bautista and Wilson, 2016*; *Fedorenko et al., 2010*; *Fedorenko et al., 2012*; *Fedorenko et al., 2020*; *Keller et al., 2001*; *Mollica et al., 2020*). We found robust bilateral responses to code problems within the MD system, a pattern that held across two very different programming languages (Python and ScratchJr), types of problems (math and string manipulation), and problem structure (sequential statements, *for* loops, and *if* statements). In contrast, responses in the language system were substantially lower than those elicited by the content-matched sentence problems and exceeded responses to the control condition (nonwords reading) only for one of the two programming languages tested.

Our work uniquely contributes to the study of computer programming in the mind and brain by addressing two core issues that made it difficult to interpret results from prior studies. First, we disentangle responses evoked by code comprehension from responses to problem content (which is often not code-specific) by contrasting code problems with content-matched sentence problems. Our findings suggest that earlier reports of left-lateralized code-evoked activity (*Siegmund et al., 2014*) may reflect processing program content rather than code comprehension per se. This distinction should also be considered when interpreting results of other studies of programming effects on brain activity, such as debugging (*Castelhano et al., 2019*), variable tracking (*Ikutani and Uwano, 2014*; *Nakagawa et al., 2014*), use of semantic cues or program layout (*Fakhoury et al., 2018*; *Siegmund et al., 2017*), program generation (*Krueger et al., 2020*), and programming expertise (*Ikutani et al., 2020*).

Second, we analyze responses in brain areas that are functionally localized in individual participants, allowing for straightforward interpretation of the observed responses (*Mather et al., 2013*; *Saxe et al., 2006*). This approach stands in contrast to the traditional approach, whereby neural responses are averaged across participants on a voxel-by-voxel basis, and the resulting activation clusters are interpreted via 'reverse inference' from anatomy (e.g., *Poldrack, 2006*; *Poldrack, 2011*). Functional localization is particularly important when analyzing responses in frontal, temporal, and parietal association cortex, which is known to be functionally heterogeneous and variable across individuals (*Blank et al., 2017*; *Braga et al., 2019*; *Fedorenko and Kanwisher, 2009*; *Frost and Goebel, 2012*; *Shashidhara et al., 2019b*; *Tahmasebi et al., 2012*; *Vázquez-Rodríguez et al., 2019*).

The results of our work align well with the results of another recent study on program comprehension (*Liu et al., 2020*). Liu et al. investigated the neural correlates of program comprehension by contrasting Python code problems with fake code. The code problem condition was similar to ours, whereas the fake code condition involved viewing scrambled code, followed by a visual recognition task. The code problems > fake code contrast is broader than ours: it includes both code comprehension (interpreting Python code) and the processing of problem content (manipulating characters in a string). Our results show that the MD system is involved in both processes, but Python code comprehension is bilateral, whereas the processing of problem content is left-lateralized. We would

therefore expect the code problems > fake code contrast to activate the MD system, engaging the left hemisphere more strongly than the right due to the demands of problem content processing. This is precisely what Liu et al. found. Further, similar to us, Liu et al. conclude that it is the MD regions, not the language regions, that are primarily involved in program comprehension.

## MD system's engagement reflects the use of domain-general resources

The fact that the MD system responds to code problems over and above content-matched sentence problems underscores the role of domain-general executive processes in code comprehension. Although cognitive processes underlying code interpretation bear parallels to logic and math tasks (*Papert, 1972*; *Pennington and Grabowski, 1990*; *Perkins and Simmons, 1988*) and to natural language comprehension/generation (*Fedorenko et al., 2019*; *Hermans and Aldewereld, 2017*), the neural activity we observe primarily resembles activity observed in response to domain-general executive tasks (*Assem et al., 2020*; *Duncan, 2010*; *Fedorenko et al., 2013*). In particular, code comprehension elicits bilateral responses within the MD system, in contrast to math and logic tasks that tend to elicit left-lateralized responses within the MD system, and in contrast to language tasks that elicit responses in the spatially and functionally distinct language system.

We found that responses in the MD system were driven both by the processing of problem content (e.g., summing the contents of an array) and by code comprehension (e.g., identifying variables referring to an array and its elements, interpreting a *for* loop, realizing that the output of the program is the variable being updated inside the *for* loop). Both of these processes plausibly require attention, working memory, inhibitory control, planning, and general flexible relational reasoning – cognitive processes long linked to the MD system (*Duncan, 2010*; *Duncan, 2013*; *Duncan and Owen, 2000*; *Miller and Cohen, 2001*) in both humans (*Assem et al., 2020*; *Shashidhara et al., 2019a*; *Woolgar et al., 2018*) and non-human primates (*Freedman et al., 2001*; *Miller et al., 1996*; *Mitchell et al., 2016*). A recent study (*Huang et al., 2019*) reported neural overlap between operations on programming data structures (which require both code comprehension and the processing of problem content) and a mental rotation task (which requires spatial reasoning). The overlap was observed within brain regions whose topography grossly resembles that of the MD system. In our study, all code-responsive brain regions outside the visual cortex also responded robustly during a spatial memory task (*Figure 3—figure supplements 3* and *4*), similar to the results reported in *Huang et al., 2019*. However, the MD system is not specifically tuned to spatial reasoning (*Duncan, 2010*; *Fedorenko et al., 2013*; *Michalka et al., 2015*), so the overlap between code comprehension and spatial reasoning likely reflects the engagement of domain-general cognitive processes, like working memory and cognitive control, as opposed to processes specific to spatial reasoning.

Furthermore, given that no regions outside of the MD system showed code-specific responses, it must be the case that code-specific knowledge representations are also *stored* within this system (see *Hasson et al., 2015*, for a general discussion of the lack of distinction between storage and computing resources in the brain). Such code-specific representations would likely include both knowledge specific to a programming language (e.g., the syntax marking an array in Java vs. Python) and knowledge of programming concepts that are shared across languages (e.g., *for* loops). Much evidence suggests that the MD system can flexibly store task-relevant information in the short term (e.g., *Fedorenko et al., 2013*; *Freedman et al., 2001*; *Shashidhara et al., 2019a*; *Wen et al., 2019*; *Woolgar et al., 2011*). However, evidence from studies on processing mathematics (e.g., *Amalric and Dehaene, 2019*) and physics (e.g., *Cetron et al., 2019*; *Fischer et al., 2016*) further suggests that the MD system can store some domain-specific representations in the long term, perhaps for evolutionarily late-emerging and ontogenetically late-acquired domains of knowledge. Our data add to this body of evidence by showing that the MD system stores and uses information required for code comprehension.

We also show that, instead of being concentrated in one region or a subset of the MD system, code-evoked responses are distributed throughout the MD system. This result seems to violate general metabolic and computational efficiency principles that govern much of the brain's architecture (*Chklovskii and Koulakov, 2004*; *Kanwisher, 2010*): if some MD neurons are, at least in part, functionally specialized to process computer code, we would expect them to be located next to each other. Three possibilities are worth considering. First, selectivity for code comprehension in a subset of the MD network may only emerge with years of experience (e.g., in professional programmers). Participants in our experiments were all proficient in the target programming language but most

had only a few years of experience with it. Second, code-selective subsets of the MD network may be detectable at higher spatial resolution, using invasive methods like electrocorticography (*Parvizi and Kastner, 2018*) or single-cell recordings (*Mukamel and Fried, 2012*). And third, perhaps the need to flexibly solve novel problems throughout one's life prevents the 'crystallization' of specialized subnetworks within the MD cortex. All that said, it may also be the case that some subset of the MD network is causally important for code comprehension even though it does not show strong selectivity for it, similar to how damage to some MD areas (mostly, in the left parietal cortex) appears to lead to deficits in numerical cognition (*Ardila and Rosselli, 2002*; *Kahn and Whitaker, 1991*; *Lemer et al., 2003*; *Rosselli and Ardila, 1989*; *Takayama et al., 1994*), even though these regions do not show selectivity for numerical tasks in fMRI (*Pinel et al., 2004*; *Shuman and Kanwisher, 2004*).

## The language system is functionally conservative

We found that the language system does not respond consistently during code comprehension in spite of numerous similarities between code and natural languages (*Fedorenko et al., 2019*). Perhaps the most salient similarity between these input types is their syntactic/combinatorial structure. Some accounts of language processing claim that syntactic operations that support language processing are highly abstract and insensitive to the nature of the to-be-combined units (e.g., *Berwick et al., 2013*; *Fitch et al., 2005*; *Fitch and Martins, 2014*; *Hauser et al., 2002*). Such accounts predict that the mechanisms supporting structure processing in language should also get engaged when we process structure in other domains, including computer code. Prior work has already put into question this idea in its broadest form: processing music, whose hierarchical structure has long been noted to have parallels with linguistic syntax (e.g., *Lerdahl and Jackendoff, 1996*; cf. *Jackendoff, 2009*), does not engage the language system (e.g., *Fedorenko et al., 2011*; *Rogalsky et al., 2011*; *Chen et al., 2020*). Our finding builds upon the results from the music domain to show that compositional input (here, variables and keywords combining into statements) and hierarchical structure (here, conditional statements and loops) do not necessarily engage language-specific regions.

Another similarity shared by computer programming and natural language is the use of symbols – units referring to concepts 'out in the world'. Studies of math and logic, domains that also make extensive use of symbols, show that those domains do not rely on the language system (*Amalric and Dehaene, 2019*; *Cohen et al., 2000*; *Fedorenko et al., 2011*; *Monti et al., 2009*; *Monti et al., 2012*; *Pinel and Dehaene, 2010*; *Varley et al., 2005*), a conclusion consistent with our findings. However, these prior results might be explained by the hypothesis that mathematics makes use of a different conceptual space altogether (*Cappelletti et al., 2001*), in which case the symbol-referent analogy would be weakened. Our work provides an even stronger test of the symbolic reference hypothesis: the computer code problems we designed are not only symbolic, but also refer to the same conceptual representations as the corresponding verbal problems (*Figure 1A*). This parallel is particularly striking in the case of ScratchJr: each code problem refers to a sequence of actions performed by a cartoon character – a clear case of reference to concepts in the physical world. And yet, the language regions do not respond to ScratchJr, showing a clear preference for language over other types of meaningful structured input (see also *Ivanova et al., 2019*).

The third similarity between code and natural language is the communicative use of those systems (*Allamanis et al., 2018*). The programming languages we chose are very high- level, meaning that they emphasize human readability (*Buse and Weimer, 2010*; *Klare, 1963*) over computational efficiency. ScratchJr is further optimized to be accessible and engaging for young children (*Sullivan and Bers, 2019*). Thus, code written in these languages is meant to be read and understood by humans, not just executed by machines. In this respect, computer code comprehension is similar to reading in natural language: the goal is to extract a meaningful message produced by another human at some point in the past. And yet the communicative nature of this activity is not sufficient to recruit the language system, consistent with previous reports showing a neural dissociation between language and other communication-related activities, such as gesture processing (*Jouravlev et al., 2019*), intentional actions (*Pritchett et al., 2018*), or theory of mind tasks (*Apperly et al., 2006*; *Dronkers et al., 1998*; *Jacoby et al., 2016*; *Paunov et al., 2019*; *Varley and Siegal, 2000*).

Of course, the lack of consistent language system engagement in code comprehension does not mean that the mechanisms underlying language and code processing are completely different. It is possible that both language and MD regions have similarly organized neural circuits that allow them to process combinatorial input or map between a symbol and the concept it refers to. However, the fact that we observed code-evoked activity primarily in the MD regions indicates that code comprehension does not load on the same neural circuits as language and needs to use domain-general MD circuits instead.

More work is required to determine why the language system showed some activity in response to Python code. The most intuitive explanation posits that the language system responds to meaningful words embedded within the code; however, this explanation seems unlikely given the fact that the responses were equally strong when reading problems with semantically meaningful identifiers (English) and semantically meaningless identifiers (Japanese; *Figure 4C*). Another possibility is that participants internally verbalized the symbols they were reading (where 'verbalize' means to retrieve the word associated with a certain symbol rather than a simple reading response, since the latter would be shared with nonwords). However, this account does not explain the fact why such verbalization would be observed for Python and not for ScratchJr, where many blocks have easy labels, such as 'jump'. It is also inconsistent with observations that even behaviors that ostensibly require subvocal rehearsal (e.g., mathematical operations) do not engage the language system (see e.g., *Amalric and Dehaene, 2019*; *Fedorenko et al., 2011*). Finally, the account that we consider most likely is that the responses were mainly driven by processing underlying problem content and thus associated with some aspect(s) of computational thinking that were more robustly present in Python compared to ScratchJr problems. Further investigations of the role of the language system in computational thinking have the potential to shed light on the exact computations supported by these regions.

Finally, it is possible that the language system may play a role in learning to program (*Prat et al., 2020*), even if it is not required to support code comprehension once the skill is learned. Studies advocating the 'coding as another language' approach (*Bers, 2019*; *Bers, 2018*; *Sullivan and Bers, 2019*) have found that treating coding as a meaning-making activity rather than merely a problem-solving skill had a positive impact on both teaching and learning to program in the classroom (*Hassenfeld et al., 2020*; *Hassenfeld and Bers, 2020*). Such results indicate that the language system and/or the general semantic system might play a role in learning to process computer code, especially in children, when the language system is still developing. This idea remains to be empirically evaluated in future studies.

## Limitations of scope

The stimuli used in our study were short and only included a few basic elements of control flow (such as *for* loops and *if* statements). Furthermore, we focused on code comprehension, which is a necessary but not sufficient component of many other programming activities, such as code generation, editing, and debugging. Future work should investigate changes in brain activity during the processing and generation of more complex code structures, such as functions, objects, and large multi-component programs. Just like narrative processing recruits systems outside the regions that support single sentence processing (*Baldassano et al., 2018*; *Blank and Fedorenko, 2020*; *Ferstl et al., 2008*; *Jacoby and Fedorenko, 2020*; *Lerner et al., 2011*; *Simony et al., 2016*), reading more complex pieces of code might recruit an extended, or a different, set of brain regions. Furthermore, as noted above, investigations of expert programmers may reveal changes in how programming knowledge and use are instantiated in the mind and brain as a function of increasing amount of domain-relevant experience.

Overall, we provide evidence that code comprehension consistently recruits the MD system – which subserves cognitive processing across multiple cognitive domains – but does not consistently engage the language system, in spite of numerous similarities between natural and programming languages. By isolating neural activity specific to code comprehension, we pave the way for future studies examining the cognitive and neural correlates of programming and contribute to the broader literature on the neural systems that support novel cognitive tools.

## Materials and methods

### Participants

For Experiment 1, we recruited 25 participants (15 women, mean age = 23.0 years, SD = 3.0). Average age at which participants started to program was 16 years (SD = 2.6); average number of years spent programming was 6.3 (SD = 3.8). In addition to Python, 20 people also reported some knowledge of Java, 18 people reported knowledge of C/C++, 4 of functional languages, and 20 of numerical languages like Matlab and R. Twenty-three participants were right-handed, one was ambidextrous, and one was left-handed (as assessed by Oldfield's [1971] handedness questionnaire); the left-handed participant had a right-lateralized language system and was excluded from the analyses, leaving 24 participants (all of whom had left-lateralized language regions, as evaluated with the language localizer task [see below]). Participants also reported their knowledge of foreign languages and completed a 1-hr-long Python proficiency test (available on the paper's website, https://github.com/ALFA-group/neural-program-comprehension).

For Experiment 2, we recruited 21 participants (13 women, mean age = 22.5 years, SD = 2.8). In addition to ScratchJr, eight people also reported some knowledge of Python, six people reported knowledge of Java, nine people reported knowledge of C/C++, one of functional languages, and fourteen of numerical languages like Matlab and R (one participant did not complete the programming questionnaire). Twenty were right-handed and one was ambidextrous; all participants had left-lateralized language regions, as evaluated with the language localizer task (see below). Two participants from Experiment 2 had to be excluded due to excessive motion during the MRI scan, leaving 19 participants.

All participants were recruited from MIT, Tufts University, and the surrounding community and paid for participation. All were native speakers of English, had normal or corrected to normal vision, and reported working knowledge of Python or ScratchJr, respectively. The sample size for both experiments was determined based on previous experiments from our group (e.g., *Blank and Fedorenko, 2020*; *Fedorenko et al., 2020*; *Ivanova et al., 2019*) and others (e.g., *Crittenden et al., 2015*; *Hugdahl et al., 2015*; *Shashidhara et al., 2019a*). The protocol for the study was approved by MIT's Committee on the Use of Humans as Experimental Subjects (COUHES). All participants gave written informed consent in accordance with protocol requirements.

### Design, materials, and procedure

All participants completed the main program comprehension task, a spatial working memory localizer task aimed at identifying the MD brain regions (*Fedorenko et al., 2011*), and a language localizer task aimed at identifying language-responsive brain regions (*Fedorenko et al., 2010*).

The program comprehension task in Experiment 1 included three conditions: programs in Python with English identifiers, programs in Python with Japanese identifiers, and sentence versions of those programs (visually presented). The full list of problems can be found on the paper's website, https://github.com/ALFA-group/neural-program-comprehension. Each participant saw 72 problems, and any given participant saw only one version of a problem. Half of the problems required performing mathematical operations, and the other half required string manipulations. In addition, both math and string-manipulation problems varied in program structure: 1/3 of the problems of each type included only sequential statements, 1/3 included a *for* loop, and 1/3 included an *if* statement.

During each trial, participants were instructed to read the problem statement and press a button when they were ready to respond (the minimum processing time was restricted to 5 s and the maximum to 50 s; mean reading time was 19 s). Once they pressed the button, four response options were revealed, and participants had to indicate their response by pressing one of four buttons on a button box. The response screen was presented for 5 s (see *Figure 1—figure supplement 1A* for a schematic of trial structure). Each run consisted of six trials (two per condition) and three fixation blocks (at the beginning and end of the run, and after the third trial), each lasting 10 s. A run lasted, on average, 176 s (SD = 34 s), and each participant completed 12 runs. Condition order was counterbalanced across runs and participants.

The program comprehension task in Experiment 2 included two conditions: short programs in ScratchJr and the sentence versions of those programs (visually presented). ScratchJr is a language

designed to teach programming concepts to young children (*Bers, 2018*): users can create events and sequences of events (stories) with a set of characters and actions. The full list of problems used in the study can be found on the paper's website. Each participant saw 24 problems, and any given participant saw only one version of a problem. Furthermore, problems varied in the complexity of the code snippet (three levels of difficulty; eight problems at each level).

During each trial, participants were presented with a fixation cross for 4 s, followed by a description (either a code snippet or a sentence) to read for 8 s. The presentation of the description was followed by 5–9 s of fixation, and then by a video (average duration: 4.13 s, SD: 1.70 s) that either did or did not match the description. Participants had to indicate whether the video matched the description by pressing one of two buttons on a button box in the scanner. The response window started with the onset of the video and included a 4 s period after the video offset. A trial lasted, on average, 27.46 s (SD = 2.54 s; see *Figure 1—figure supplement 1B*, for a schematic of trial structure). Each run consisted of six trials (three per condition), and a 10 s fixation at the beginning and end of the run. A run lasted, on average, 184.75 s (SD = 3.86 s); each participant completed four runs. Condition order was counterbalanced across runs and participants.

The spatial working memory task was conducted in order to identify the MD system within individual participants. Participants had to keep track of four (easy condition) or eight (hard condition) sequentially presented locations in a 3 × 4 grid (*Figure 1B*; *Fedorenko et al., 2011*). In both conditions, they performed a two-alternative forced-choice task at the end of each trial to indicate the set of locations they just saw. The hard >easy contrast has been previously shown to reliably activate bilateral frontal and parietal MD regions (*Assem et al., 2020*; *Blank et al., 2014*; *Fedorenko et al., 2013*). Numerous studies have shown that the same brain regions are activated by diverse executively demanding tasks (*Duncan and Owen, 2000*; *Fedorenko et al., 2013*; *Hugdahl et al., 2015*; *Shashidhara et al., 2019a*; *Woolgar et al., 2011*). Stimuli were presented in the center of the screen across four steps. Each step lasted 1 s and revealed one location on the grid in the easy condition, and two locations in the hard condition. Each stimulus was followed by a choice-selection step, which showed two grids side by side. One grid contained the locations shown across the previous four steps, while the other contained an incorrect set of locations. Participants were asked to press one of two buttons to choose the grid that showed the correct locations. Condition order was counterbalanced across runs. Experimental blocks lasted 32 s (with four trials per block), and fixation blocks lasted 16 s. Each run (consisting of four fixation blocks and 12 experimental blocks) lasted 448 s. Each participant completed two runs.

The language localizer task was conducted in order to identify the language system within individual participants. Participants read sentences (e.g., NOBODY COULD HAVE PREDICTED THE EARTHQUAKE IN THIS PART OF THE COUNTRY) and lists of unconnected, pronounceable nonwords (e.g., U BIZBY ACWORRILY MIDARAL MAPE LAS POME U TRINT WEPS WIBRON PUZ) in a blocked design. Each stimulus consisted of twelve words/nonwords. For details of how the language materials were constructed, see *Fedorenko et al., 2010*. The materials are available at http://web.mit.edu/evelina9/www/funcloc/funcloc_localizers.html. The sentences > nonword lists contrast isolates processes related to language comprehension (responses evoked by, e.g., visual perception and reading are subtracted out) and has been previously shown to reliably activate left-lateralized fronto-temporal language processing regions, be robust to changes in task and materials, and activate the same regions regardless of whether the materials were presented visually or auditorily (*Fedorenko et al., 2010*; *Mahowald and Fedorenko, 2016*; *Scott et al., 2017*). Further, a similar network emerges from task-free resting-state data (*Braga et al., 2020*). Stimuli were presented in the center of the screen, one word/nonword at a time, at the rate of 450 ms per word/nonword. Each stimulus was preceded by a 100 ms blank screen and followed by a 400 ms screen showing a picture of a finger pressing a button, and a blank screen for another 100 ms, for a total trial duration of 6 s. Participants were asked to press a button whenever they saw the picture of a finger pressing a button. This task was included to help participants stay alert. Condition order was counterbalanced across runs. Experimental blocks lasted 18 s (with three trials per block), and fixation blocks lasted 14 s. Each run (consisting of 5 fixation blocks and 16 experimental blocks) lasted 358 s. Each participant completed two runs.

### fMRI data acquisition

Structural and functional data were collected on the whole-body, 3 Tesla, Siemens Trio scanner with a 32-channel head coil, at the Athinoula A. Martinos Imaging Center at the McGovern Institute for Brain Research at MIT. T1-weighted structural images were collected in 176 sagittal slices with 1 mm isotropic voxels (TR = 2,530 ms, TE = 3.48 ms). Functional, blood oxygenation level dependent (BOLD), data were acquired using an EPI sequence (with a 90° flip angle and using GRAPPA with an acceleration factor of 2), with the following acquisition parameters: thirty-one 4 mm thick near-axial slices acquired in the interleaved order (with 10% distance factor), 2.1 mm × 2.1 mm in-plane resolution, FoV in the phase encoding (A >> P) direction 200 mm and matrix size 96 mm × 96 mm, TR = 2,000 ms and TE = 30 ms. The first 10 s of each run were excluded to allow for steady state magnetization.

### fMRI data preprocessing

MRI data were analyzed using SPM12 and custom MATLAB scripts (available in the form of an SPM toolbox from http://www.nitrc.org/projects/spm_ss). Each participant's data were motion corrected and then normalized into a common brain space (the Montreal Neurological Institute [MNI] template) and resampled into 2 mm isotropic voxels. The data were then smoothed with a 4 mm FWHM Gaussian filter and high-pass filtered (at 128 s). Effects were estimated using a General Linear Model (GLM) in which each experimental condition was modeled with a boxcar function convolved with the canonical hemodynamic response function (HRF). For the localizer experiments, we modeled the entire blocks. For the Python program comprehension experiment, we modeled the period from the onset of the code/sentence problem and until the button press (the responses were modeled as a separate condition; see *Figure 1—figure supplement 1A*); for the ScratchJr program comprehension experiment, we modeled the period of the code/sentence presentation (the video and the response were modeled as a separate condition; see *Figure 1—figure supplement 1B*).

### Defining MD and language functional regions of interest (fROIs)

The fROI analyses examined responses in individually defined MD and language fROIs. These fROIs were defined using the group-constrained subject-specific (GSS) approach (*Fedorenko et al., 2010*; *Julian et al., 2012*) where a set of spatial masks, or parcels, is combined with each individual subject's localizer activation map, to constrain the definition of individual fROIs. The parcels delineate the expected gross locations of activations for a given contrast based on prior work and large numbers of participants and are sufficiently large to encompass the variability in the locations of individual activations. For the MD system, we used a set of 20 parcels (10 in each hemisphere) derived from a group-level probabilistic activation overlap map for the hard >easy spatial working memory contrast in 197 participants. The parcels included regions in frontal and parietal lobes, as well as a region in the anterior cingulate cortex. For the language system, we used a set of six parcels derived from a group-level probabilistic activation overlap map for the sentences > nonwords contrast in 220 participants. The parcels included two regions in the left inferior frontal gyrus (LIFG, LIFGorb), one in the left middle frontal gyrus (LMFG), two in the left temporal lobe (LAntTemp and LPostTemp), and one extending into the angular gyrus (LAngG). Both sets of parcels are available on the paper's website; see *Figure 3—figure supplement 1* for labeled images of MD and language parcels. Within each parcel, we selected the top 10% most localizer-responsive voxels, based on the *t*-values (see, e.g., Figure 1 in *Blank et al., 2014*, or *Shain et al., 2020* for sample MD and language fROIs). Individual fROIs defined this way were then used for subsequent analyses that examined responses to code comprehension.

### Examining the functional response profiles of the MD and language fROIs

#### Univariate analyses

We evaluated MD and language system responses by estimating their response magnitudes to the conditions of interest using individually defined fROIs (see above). For each fROI in each participant, we averaged the responses across voxels to get a single value for each of the conditions (the responses to the localizer conditions were estimated using an across-runs cross-validation procedure, where one run was used to define the fROI and the other to estimate the response

magnitudes, then the procedure was repeated switching which run was used for fROI definition vs. response estimation, and finally the estimates were averaged to derive a single value per condition per fROI per participant). We then ran a linear mixed-effect regression model to compare the responses to the critical code problem condition with (a) the responses to the sentence problem condition from the critical task, and (b) the responses to the nonword reading condition from the language localizer task. We included *condition* as a fixed effect and *participant* and *fROI* as random intercepts. For the MD system, we additionally tested the main (fixed) effect of *hemisphere* and the interaction between *hemisphere* and *condition*. We used dummy coding for *condition*, with code problems as the reference category, and sum coding for *hemisphere*. For follow-up analyses, we used the variable of interest (*problem type/structure/identifier language*) as a fixed effect and *participant* and *fROI* as random intercepts; dummy coding was used for all variables of interest. For fROI analyses, we used *condition* as a fixed effect and *participant* as a random intercept. The analyses were run using the *lmer* function from the *lme4* R package (*Bates et al., 2015*); statistical significance of the effects was evaluated using the *lmerTest* package (*Kuznetsova et al., 2017*).

### Spatial correlation analyses

To further examine the similarity of the fine-grained patterns of activation between conditions in the language system, we calculated voxel-wise spatial correlations in activation magnitudes within the code problem condition (between odd and even runs), within the sentence problem condition (between odd and even runs), between these two conditions (we used odd and even run splits here, too, to match the amount of data for the within- vs. between-condition comparisons, and averaged the correlation values across the different splits), and between these two critical conditions and each of the sentence and nonword reading conditions from the language localizer. The correlation values were calculated for voxels in each participant's language fROIs, and then averaged across participants and fROIs for plotting (the values were weighted by fROI size). We also used the *lme4* R package to calculate statistical differences between spatial correlation values for code vs. other conditions (with *participant* and *fROI* as random intercepts); for this analysis, the correlation values were Fischer-transformed.

## Whole-brain analyses

For each of the critical experiments (Python and ScratchJr), we conducted (a) the GSS analysis (*Fedorenko et al., 2010*; *Julian et al., 2012*), and (b) the traditional random effects group analysis (*Holmes and Friston, 1998*) using the code problems > sentence problems contrast. The analyses were performed using the spm_ss toolbox (http://www.nitrc.org/projects/spm_ss), which interfaces with SPM and the CONN toolbox (https://www.nitrc.org/projects/conn).

## Acknowledgements

We would like to acknowledge the Athinoula A Martinos Imaging Center at the McGovern Institute for Brain Research at MIT and its support team (Steve Shannon, Atsushi Takahashi, and Dima Ayyash), Rachel Ryskin for advice on statistics, Alfonso Nieto-Castañón for help with analyses, ALFA group at CSAIL for helpful discussions on the experiment design, and Josef Affourtit, Yev Diachek, and Matt Siegelman (EvLab), and Ruthi Aladjem, Claudia Mihm, and Kaitlyn Leidl (DevTech research group at Tufts University) for technical support during experiment design and data collection.

---

## Additional information

### Funding

| Funder | Grant reference number | Author |
|--------|------------------------|--------|
| National Science Foundation | #1744809 | Marina U Bers<br>Evelina Fedorenko |
| Department of Brain and Cognitive Science, MIT | | Evelina Fedorenko |
| McGovernInstitute for Brain | | Evelina Fedorenko |

Research

The funders had no role in study design, data collection and interpretation, or the decision to submit the work for publication.

## Author contributions
Anna A Ivanova, Conceptualization, Data curation, Software, Formal analysis, Validation, Investigation, Visualization, Methodology, Writing - original draft, Project administration; Shashank Srikant, Conceptualization, Software, Investigation, Methodology, Writing - review and editing; Yotaro Sueoka, Data curation, Software, Investigation, Methodology; Hope H Kean, Data curation, Software, Formal analysis, Investigation, Methodology; Riva Dhamala, Software, Investigation, Methodology; Una-May O'Reilly, Conceptualization, Supervision, Project administration, Writing - review and editing; Marina U Bers, Evelina Fedorenko, Conceptualization, Resources, Supervision, Funding acquisition, Methodology, Project administration, Writing - review and editing

## Author ORCIDs
Anna A Ivanova ⬛ https://orcid.org/0000-0002-1184-8299
Yotaro Sueoka ⬛ https://orcid.org/0000-0001-6417-9052
Marina U Bers ⬛ https://orcid.org/0000-0003-0206-1846
Evelina Fedorenko ⬛ https://orcid.org/0000-0003-3823-514X

## Ethics
Human subjects: MIT's Committee on the Use of Humans as Experimental Subjects (COUHES) approved the protocol for the current study (protocol #0907003336R010, "fMRI Investigations of Language and its Relationship to Other Cognitive Abilities"). All participants gave written informed consent in accordance with protocol requirements.

## Decision letter and Author response
Decision letter https://doi.org/10.7554/eLife.58906.sa1
Author response https://doi.org/10.7554/eLife.58906.sa2

# Additional files

## Supplementary files
• Supplementary file 1. Statistical analysis of functional ROIs in the multiple demand system. Table 1 – Experiment 1 (Python); Table 2 – Experiment 2 (ScratchJr).

• Transparent reporting form

## Data availability
Materials used for the programming tasks, fROI responses in individual participants (used for generating Figures 2-4), behavioral data, and analysis code files are available on the paper's website https://github.com/ALFA-group/neural-program-comprehension (copy archived at https://archive.softwareheritage.org/swh:1:rev:616e893d05038da620bdf9f2964bd3befba75dc5/). Whole brain activation maps are available at https://osf.io/9jfn5/.

The following dataset was generated:

| Author(s) | Year | Dataset title | Dataset URL | Database and Identifier |
|---|---|---|---|---|
| Ivanova AA, Srikant S, Sueoka Y, Kean HH, Dhamala R, O'Reilly U-M, Bers MU, Fedorenko E | 2020 | Comprehension of computer code relies primarily on domain-general executive resources | https://osf.io/9jfn5/ | Open Science Framework, 10.17605/OSF.IO/9JFN5 |

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
