## [Decision Letter]

**Decision letter after peer review:**

Thank you for submitting your article "Comprehension of computer code relies primarily on domain-general executive resources" for consideration by *eLife*. Your article has been reviewed by two peer reviewers, and the evaluation has been overseen by a Reviewing Editor and Timothy Behrens as the Senior Editor. The following individuals involved in review of your submission have agreed to reveal their identity: William Matchin (Reviewer #1); Ina Bornkessel-Schlesewsky (Reviewer #2).

The reviewers have discussed the reviews with one another and the Reviewing Editor has drafted this decision to help you prepare a revised submission.

First, thank you for taking part in the review process.

As you know, *eLife* is invested in changing scientific publishing and experimenting to embody that change, even if that involves a degree of risk in order to find workable changes. In this spirit, the remit of the co-submission format is to ask if the scientific community is enriched by the data presented in the co-submitted manuscripts together more so than it would be by the papers apart, or if only one paper was presented to the community. In other words, are the conclusions that can be made are stronger or clearer when the manuscripts are considered together rather than separately? We felt that despite significant concerns with each paper individually, especially regarding the theoretical structures in which the experimental results could be interpreted, that this was the case.

We want to be very clear that in a non-co-submission case we would have substantial and serious concerns about the interpretability and robustness of the Liu et al. submission given its small sample size. Furthermore, the reviewers' concerns about the suitability of the control task differed substantially between the manuscripts. We share these concerns. However, despite these differences in control task and sample size, the Liu et al. and Ivanova et al. submissions nonetheless replicated each other – the language network was not implicated in processing programming code. The replication substantially mitigates the concerns shared by us and the reviewers about sample size and control tasks. The fact that different control tasks and sample sizes did not change the overall pattern of results, in our view, is affirmation of the robustness of the findings, and the value that both submissions presented together can offer the literature.

In sum, there were concerns that both submissions were exploratory in nature, lacking a strong theoretical focus, and relied on functional localizers on novel tasks. However, these concerns were mitigated by the following strengths. Both tasks ask a clear and interesting question. The results replicate each other despite task differences. In this way, the two papers strengthen each other. Specifically, the major concerns for each paper individually are ameliorated when considering them as a whole.

In your revisions, please address the concerns of the reviewers, including, specifically, the limits of interpretation of your results with regard to control task choice, the discussion of relevant literature mentioned by the reviewers, and most crucially, please contextualize your results with regard to the other submission's results.

Reviewer #1:

The manuscript is well-written and the methods are clear and rigorous, representing a clear advance on previous research comparing computer code programming to language. The conclusions with respect to which brain networks computer programming activates are compelling and well conveyed. This paper is useful to the extent that the conclusions are focused on the empirical findings: whether or not code activates language-related brain regions (answer: no). However, the authors appear to be also testing whether or not any of the mechanisms involved in language are recruited for computer programming. The problem with this goal is that the authors do not present or review a theory of the representations and mechanisms involved in computer programming, as has been developed for language (e.g. Adger, 2013; Bresnan, 2001; Chomsky, 1965, 1981, 1995; Goldberg, 1995; Hornstein, 2009; Jackendoff, 2002; Levelt, 1989; Lewis and Vasishth, 2005; Vosse and Kempen, 2000).

1) "The fact that coding can be learned in adulthood suggests that it may rely on existing cognitive systems." "Finally, code comprehension may rely on the system that supports comprehension of natural languages: to successfully process both natural and computer languages, we need to access stored meanings of words/tokens and combine them using hierarchical syntactic rules (Fedorenko et al., 2019; Murnane, 1993; Papert, 1993) – a similarity that, in theory, should make the language circuits well-suited for processing computer code." If we understand stored elements and computational structure in the broadest way possible without breaking this down more, many domains of cognition would be shared in this way. The authors should illustrate in more detail how the psychological structure of computer programming parallels language. Is there an example of hierarchical structure in computer code? What is the meaning of a variable/function in code, and how does this compare to meaning in language?

2) "Our findings, along with prior findings from math and logic (Amalric and Dehaene, 2019; Monti et al., 2009, 2012), argue against this possibility: the language system does not respond to meaningful structured input that is non-linguistic." This is an overly simple characterization of the word "meaningful". The meaning of math and logic are not the same as in language. Both mathematics and computer programming have logical structure to them, but the nature of this structure and the elements that are combined in language are different. Linguistic computations take as input complex atoms of computation that have phonological and conceptual properties. These atoms are commonly used to refer to entities "in the world" with complex semantic properties and often have rich associated imagery. Linguistic computations output complex, monotonically enhanced forms. So cute + dogs = cute dogs, chased + cute dogs = chased cute dogs, etc. This is very much unlike mathematics and computer programming, where we typically do not make reference to the "real world" using these expressions to interlocuters, and outputs of an expression are not monotonic, structure-preserving combinations of the input elements, and there is no semantic enhancement that occurs through increased computation. This bears much more discussion in the paper, if the authors intend to make claims regarding shared/distinct computations between computer programming and language.

3) More importantly, even if there were shared mechanisms between computer code programming and language, I'm not sure we can use reverse inference to strongly test this hypothesis. As Poldrack, 2006, pointed out, reverse inference is sharply limited by the extent to which we know how cognition maps onto the brain. This is a similar point to Poeppel and Embick, 2005, who pointed out that different mechanisms of language could be implemented in the brain in a large variety of ways, only one of which is big pieces of cortical tissue. In this sense, there could in fact be shared mechanisms between language and code (e.g. oscillatory dynamics, connectivity patterns, subcortical structures), but these mechanisms might not be aligned with the cortical territory associated with language-related brain regions. The authors should spend much additional time discussing these alternative possibilities.

Reviewer #2:

This carefully designed fMRI study examines an interesting question, namely how computer code – as a "cognitive/cultural invention" – is processed by the human brain. The study has a number of strengths, including: use of two very different programming languages (Python and Scratch Jr.) in two experiments; direct comparison between code problems and "content-matched sentence problems" to disentangle code comprehension from problem content; control for the impact of lexical information in code passages by replacing variable names with Japanese translations; and consideration of inter-individual differences in programming proficiency. I do, however, have some questions regarding the interpretation of the results in mechanistic terms, as detailed below.

1) Code comprehension versus underlying problem content

I am generally somewhat sceptical in regard to the use of functional localisers in view of the assumptions that necessarily enter into the definition of a localiser task. In addition, an overlap between the networks supporting two different tasks doesn't imply comparable neural processing mechanisms. With the present study, however, I was impressed by the authors' overall methodological approach. In particular, I found the supplementation of the localiser-based approach with the comparison between code problems and analogous sentence problems rather convincing.

However, while I agree that computational thinking does not require coding / code comprehension, it is less clear to me what code comprehension involves when it is stripped of the computational thinking aspect. Knowing how to approach a problem algorithmically strikes me as a central aspect of coding. What, then, is being measured by the code problem versus sentence problem comparison? Knowledge of how to implement a certain computational solution within a particular programming language? The authors touch upon this briefly in the Discussion section of the paper, but I am not fully convinced by their arguments. Specifically, they state:

"The process of code comprehension includes retrieving code-related knowledge from memory and applying it to the problems at hand. This application of task-relevant knowledge plausibly requires attention, working memory, inhibitory control, planning, and general flexible reasoning-cognitive processes long linked to the MD system […].”

Shouldn't all of this also apply (or even apply more strongly) to processing of the underlying problem content rather than to code comprehension per se?

According to the authors, the extent to which code-comprehension-related activity reflects problem content varies between different systems. They conclude that "MD responses to code […] do not exclusively reflect responses to problem content", while they argue on the basis of their voxel-wise correlation analysis that "the language system's response to code is largely (although not completely) driven by problem content. However, unless I have missed something, the latter analysis was only undertaken for the language system but not for the other systems under examination. Was there a particular reason for this? Also, what are the implications of observing problem content-driven responses within the language system for the authors' conclusion that this system is "functionally conservative"?

Overall, the paper would be strengthened by more clarity in regard to these issues – and specifically a more detailed discussion of what code comprehension may amount to in mechanistic terms when it is stripped of computational thinking.

2) Implications of using reading for the language localiser task

Given that reading is also a cultural invention, is it really fair to say that coding is being compared to the "language system" here rather than to the "reading system" (in view of the visual presentation of the language task)? The possible implications of this for the interpretation of the data should be considered.

3) Possible effects of verbalisation?

It appears possible that participants may have internally verbalised code problems – at least to a certain extent (and likely with a considerable degree of inter-individual variability). How might this have affected the results of the present study? Could verbalisation be related to the highly correlated response between code problems and language problems within the language system?

---

## [Author Response]

[…] In your revisions, please address the concerns of the reviewers, including, specifically, the limits of interpretation of your results with regard to control task choice, the discussion of relevant literature mentioned by the reviewers, and most crucially, please contextualize your results with regard to the other submission's results.

We thank the editor and the reviewers for their thoughtful evaluation. We have now addressed the reviewers’ comments (see below) and added a paragraph comparing our results with those of Liu et al.:

“The results of our work align well with the results of another recent study on program comprehension (Liu et al., 2020). […] This is precisely what Liu et al. find. Further, similar to us, Liu et al. conclude that it is the MD regions, not the language regions, that are primarily involved in program comprehension.”

We also added a reference to Liu et al.’s work in the Introduction:

“However, none of these prior studies sought to explicitly distinguish code comprehension from other programming-related processes, and none of them provide quantitative evaluations of putative shared responses to code and other tasks, such as working memory, math, or language (cf. Liu et al., 2020; see Discussion).”

Additionally, we have adjusted the statistical results in order to fix two minor bugs in our code (accidentally excluding results from one participant and using dummy coding instead of sum coding for Hemisphere when analyzing MD system activity). The resulting changes were numerically small and did not affect any of our conclusions. We also formatted supplementary figure/file references in accordance with the journal’s requirements.

Reviewer #1:The manuscript is well-written and the methods are clear and rigorous, representing a clear advance on previous research comparing computer code programming to language. The conclusions with respect to which brain networks computer programming activates are compelling and well conveyed. This paper is useful to the extent that the conclusions are focused on the empirical findings: whether or not code activates language-related brain regions (answer: no). However, the authors appear to be also testing whether or not any of the mechanisms involved in language are recruited for computer programming. The problem with this goal is that the authors do not present or review a theory of the representations and mechanisms involved in computer programming, as has been developed for language (e.g. Adger, 2013; Bresnan, 2001; Chomsky, 1965, 1981, 1995; Goldberg, 1995; Hornstein, 2009; Jackendoff, 2002; Levelt, 1989; Lewis and Vasishth, 2005; Vosse and Kempen, 2000).

Thank you for the positive evaluation! We agree that the main value of our paper is examining the contributions of the two brain networks to computer programming. Unfortunately, the theories of representations/computations involved in computer code comprehension are quite underdeveloped and have very little experimental support. Thus, the main mechanistic distinction we investigate in this work is a high-level distinction between code comprehension and the processing of problem content. We have adjusted the framing and the title of our manuscript to tone down and/or clarify our theoretical stance (see responses to specific points below).

1) "The fact that coding can be learned in adulthood suggests that it may rely on existing cognitive systems." "Finally, code comprehension may rely on the system that supports comprehension of natural languages: to successfully process both natural and computer languages, we need to access stored meanings of words/tokens and combine them using hierarchical syntactic rules (Fedorenko et al., 2019; Murnane, 1993; Papert, 1993) – a similarity that, in theory, should make the language circuits well-suited for processing computer code." If we understand stored elements and computational structure in the broadest way possible without breaking this down more, many domains of cognition would be shared in this way. The authors should illustrate in more detail how the psychological structure of computer programming parallels language. Is there an example of hierarchical structure in computer code? What is the meaning of a variable/function in code, and how does this compare to meaning in language?

We have modified the sections to further illustrate potential structural similarities between language and code:

“Finally, code comprehension may rely on the system that supports comprehension of natural languages (Fedorenko et al., 2019; Murnane, 1993; Papert, 1993). Like language, computer code makes heavy use of hierarchical structures (e.g., loops, conditionals, and recursive statements), and, like language, it can convey an unlimited amount of meaningful information (e.g., describing objects or action sequences). These similarities could, in principle, make the language circuits well-suited for processing computer code.”

“Such accounts predict that the mechanisms supporting structure processing in language should also get engaged when we process structure in other domains, including computer code. […] Our finding builds upon these results to show that compositional input (here, variables and keywords combining into statements) and hierarchical structure (here, conditional statements and loops) do not necessarily engage language-specific regions.”

We have also added more details about meaning in language vs. code (see point 2).

2) "Our findings, along with prior findings from math and logic (Amalric and Dehaene, 2019; Monti et al., 2009, 2012), argue against this possibility: the language system does not respond to meaningful structured input that is non-linguistic." This is an overly simple characterization of the word "meaningful". The meaning of math and logic are not the same as in language. Both mathematics and computer programming have logical structure to them, but the nature of this structure and the elements that are combined in language are different. Linguistic computations take as input complex atoms of computation that have phonological and conceptual properties. These atoms are commonly used to refer to entities "in the world" with complex semantic properties and often have rich associated imagery. Linguistic computations output complex, monotonically enhanced forms. So cute + dogs = cute dogs, chased + cute dogs = chased cute dogs, etc. This is very much unlike mathematics and computer programming, where we typically do not make reference to the "real world" using these expressions to interlocuters, and outputs of an expression are not monotonic, structure-preserving combinations of the input elements, and there is no semantic enhancement that occurs through increased computation. This bears much more discussion in the paper, if the authors intend to make claims regarding shared/distinct computations between computer programming and language.

We thank the reviewer for raising this interesting and important point. We agree that it is important to clarify what we mean by “meaning”. We therefore significantly expanded the section in question to clarify our statement:

“Another similarity shared by computer programming and natural language is the use of symbols – units referring to concepts “out in the world”. […] And yet the communicative nature of this activity is not sufficient to recruit the language system, consistent with previous reports showing a neural dissociation between language and gesture processing (Jouravlev et al., 2019), intentional actions (Pritchett et al., 2018) or theory of mind tasks (Apperly et al., 2006; Dronkers et al., 1998; Jacoby et al., 2016; Paunov et al., 2019; Varley and Siegal, 2000).”

Note that this section addresses the comments about reference to external entities and the communicative intent (which, as we state above, we think are partially shared between language and code). The paragraph preceding these (see point 1) also addresses the point about compositionality, at least in its most basic definition. We agree that language has both semantic and compositional properties that make it “special” (rich associations, imagery, semantic enhancement), all of which might account for the high functional selectivity of the language regions. However, we think that the discussion of such possibly unique properties of language is outside the scope of this paper and therefore limit ourselves to listing the putative *shared* properties of language and code.

3) More importantly, even if there were shared mechanisms between computer code programming and language, I'm not sure we can use reverse inference to strongly test this hypothesis. As Poldrack, 2006, pointed out, reverse inference is sharply limited by the extent to which we know how cognition maps onto the brain. This is a similar point to Poeppel and Embick, 2005, who pointed out that different mechanisms of language could be implemented in the brain in a large variety of ways, only one of which is big pieces of cortical tissue. In this sense, there could in fact be shared mechanisms between language and code (e.g. oscillatory dynamics, connectivity patterns, subcortical structures), but these mechanisms might not be aligned with the cortical territory associated with language-related brain regions. The authors should spend much additional time discussing these alternative possibilities.

The reviewer is right in that we here focus on one way in which two cognitive functions might share resources (through cortical overlap; though note that our whole-brain analyses rule out overlap at the subcortical level, too). It is absolutely true that functionally distinct cortical areas could still (a) implement similar computations, and/or (b) interact, e.g., via oscillatory dynamics. We have addressed this point in a separate paragraph.

“Of course, the lack of consistent language system engagement in code comprehension does not mean that the mechanisms underlying language and code processing are completely different. […] However, the fact that we observed code-evoked activity primarily in the MD regions indicates that code comprehension does not load on the same neural circuits as language and needs to use domain-general MD circuits instead.”

Reviewer #2:[…] I do, however, have some questions regarding the interpretation of the results in mechanistic terms, as detailed below.1) Code comprehension versus underlying problem contentI am generally somewhat sceptical in regard to the use of functional localisers in view of the assumptions that necessarily enter into the definition of a localiser task. In addition, an overlap between the networks supporting two different tasks doesn't imply comparable neural processing mechanisms. With the present study, however, I was impressed by the authors' overall methodological approach. In particular, I found the supplementation of the localiser-based approach with the comparison between code problems and analogous sentence problems rather convincing.However, while I agree that computational thinking does not require coding / code comprehension, it is less clear to me what code comprehension involves when it is stripped of the computational thinking aspect. Knowing how to approach a problem algorithmically strikes me as a central aspect of coding. What, then, is being measured by the code problem versus sentence problem comparison? Knowledge of how to implement a certain computational solution within a particular programming language?

The reviewer is right – it will be helpful to provide further clarifications of what it means to process problem content vs. process code in the Discussion section (in addition to the definition we provide in the Introduction). We address the specific points below. Additionally, prompted by the comments from reviewer 1, we added some more information about the putative cognitive processes underlying code comprehension and their possible parallels with language processing (see reviewer 1, points 1 and 2).

The authors touch upon this briefly in the Discussion section of the paper, but I am not fully convinced by their arguments. Specifically, they state:"The process of code comprehension includes retrieving code-related knowledge from memory and applying it to the problems at hand. This application of task-relevant knowledge plausibly requires attention, working memory, inhibitory control, planning, and general flexible reasoning-cognitive processes long linked to the MD system […].Shouldn't all of this also apply (or even apply more strongly) to processing of the underlying problem content rather than to code comprehension per se?

This paragraph indeed applies to both processes. We have adjusted it below, including examples of processing code vs. problem content:

“We found that responses in the MD system were driven both by the processing of problem content (e.g., summing the contents of an array) and by code comprehension (e.g., identifying variables referring to an array and its elements, interpreting a for-loop, realizing that the output of the program is the variable being updated inside the for-loop). […] The overlap was observed within brain regions whose topography grossly resembles that of the MD system.”

We also added a sentence about processes specific to code comprehension to the next paragraph:

“Furthermore, given that no regions outside of the MD system showed code-specific responses, it must be the case that code-specific knowledge representations are also *stored* within this system (see Hasson et al., 2015, for a general discussion of the lack of distinction between storage and computing resources in the brain). Such code-specific representations would likely include both knowledge specific to a programming language (e.g. the syntax marking an array in Java vs. Python) and knowledge of programming concepts that are shared across languages (e.g. *for* loops).”

According to the authors, the extent to which code-comprehension-related activity reflects problem content varies between different systems. They conclude that "MD responses to code […] do not exclusively reflect responses to problem content", while they argue on the basis of their voxel-wise correlation analysis that "the language system's response to code is largely (although not completely) driven by problem content. However, unless I have missed something, the latter analysis was only undertaken for the language system but not for the other systems under examination. Was there a particular reason for this?

The reason is that, for the MD system, we can make this inference based on the univariate analyses alone, but for the language system we cannot. To clarify this point, we have added an explanation to the section motivating the spatial correlation analysis and clarified our conclusion to highlight that both MD and language systems are sensitive to problem content. Finally, for completeness, we have also added a supplementary figure showing the spatial correlation plot for the MD system.

“Finally, we investigated whether the responses to Python code problems within the language system were driven by code comprehension specifically or rather by the underlying problem content. […] Thus, in both MD and language systems, response to Python code is driven both by problem content and by code-specific responses.

Overall, we found that the language system responded to code problems written in Python but not in ScratchJr. Furthermore, Python responses were driven not only by code comprehension, but also by the processing of problem content. We conclude that successful comprehension of computer code can proceed without engaging the language network.”

Also, what are the implications of observing problem content-driven responses within the language system for the authors' conclusion that this system is "functionally conservative"?

We thank the reviewer for drawing our attention to the fact that we do not explicitly discuss the possible explanations of the language system’s responses to Python code. We have added a paragraph to the Discussion section to fill this gap. It mentions both the content-driven responses and possible verbalization confounds (addressing point 3):

“More work is required to determine why the language system showed some activity in response to Python code. […] Further investigations of the role of the language system in computational thinking have the potential to shed light on the exact computations supported by these regions.”

Overall, the paper would be strengthened by more clarity in regard to these issues – and specifically a more detailed discussion of what code comprehension may amount to in mechanistic terms when it is stripped of computational thinking.2) Implications of using reading for the language localiser taskGiven that reading is also a cultural invention, is it really fair to say that coding is being compared to the "language system" here rather than to the "reading system" (in view of the visual presentation of the language task)? The possible implications of this for the interpretation of the data should be considered.

We believe that we are examining the language system rather than the reading system because (a) we are restricting our analysis to left hemisphere fronto-temporal parcels that encompass language-related activity (cf. the ventral visual areas that specifically support reading; Baker et al., 2007), (b) we are defining the fROIs based on the sentences vs. pronounceable nonwords contrast (both of these conditions involve reading, but only sentences have linguistic structure + meaning). Moreover, these regions have been previously shown to respond to both spoken and written language (Deniz et al., 2019; Fedorenko et al., 2010; Nakai et al., 2020; Regev et al., 2013; Scott et al., 2017), and damage to these regions leads to linguistic difficulties in both reading and listening (for comprehension), and writing and speaking (for production).

To clarify, we have added more information about the language system in the main text:

“These regions respond robustly to linguistic input, both visual and auditory (Deniz et al., 2019; Fedorenko et al., 2010; Nakai et al., 2020; Regev et al., 2013; Scott et al., 2017). However, they show little or no response to tasks in non-linguistic domains”

“The sentences > nonword-lists contrast isolates processes related to language comprehension (responses evoked by, e.g., visual perception and reading are subtracted out) and has been previously shown to reliably activate left-lateralized fronto-temporal language processing regions, be robust to changes in task and materials, and activate the same regions regardless of whether the materials were presented visually or auditorily (Fedorenko et al., 2010; Mahowald and Fedorenko, 2016; Scott et al., 2017).”

3) Possible effects of verbalisation?It appears possible that participants may have internally verbalised code problems – at least to a certain extent (and likely with a considerable degree of inter-individual variability). How might this have affected the results of the present study? Could verbalisation be related to the highly correlated response between code problems and language problems within the language system?

Internal verbalization is indeed an important confound to keep in mind when trying to dissociate the neural correlates of language and other cognitive functions. It is not a major threat to claims that the language system is *not* engaged in code comprehension, so our main conclusion is not affected by this potential confound. However, the reviewer is right in pointing out that verbalization may underlie the language system’s responses to Python code. While this is possible, we think such an explanation is unlikely, since ScratchJr problems would have been even easier to verbalize, and yet they do not evoke language responses. We have included a section addressing verbalization in the Discussion; this text is also included in the response to point (1).

“More work is required to determine why the language system showed some activity in response to Python code. […] It is also inconsistent with observations that even behaviors that ostensibly require subvocal rehearsal (e.g., mathematical operations) do not engage the language system (see e.g., Amalric and Dehaene, 2019; Fedorenko et al., 2011).”